# Perturbation-based trunk stabilization training in elite rowers: A pilot study

**Robin Schäfer** [ID] *, **Hendrik Schäfer** [ID], **Petra Platen**

Department of Sports Medicine and Sports Nutrition, Faculty of Sport Science, Ruhr University Bochum, Bochum, Germany

* robin.schaefer@rub.de

## Abstract

### Introduction

Low back pain is a major health issue in elite rowers. High training volume, frequent flexion movements of the lower spine and rotational movement in sweep rowing contribute to increased spinal strain and neuropathological patterns. Perturbation-based trunk stabilization training (PTT) may be effective to treat neuromuscular deficits and low back pain.

### Methods

All boat classes (8+, 4+/-, 2-) of the male German national sweep rowing team participated in this non-randomized parallel group study. We included 26 athletes (PTT: n = 12, control group: n = 14) in our analysis. Physical and Sports therapists conducted 16 individualized PTT sessions á 30–40 minutes in 10 weeks, while the control group kept the usual routines. We collected data before and after intervention on back pain intensity and disability, maximum isometric trunk extension and flexion, jump height and postural sway of single-leg stance.

### Results

We found less disability (5.3 points, 95% CI [0.4, 10.1], g = 0.42) for PTT compared to control. Pain intensity decreased similar in both groups (-14.4 and -15.4 points), yielding an inconclusive between-group effect (95% CI [-16.3, 14.3]). Postural sway, strength and jump height tend to have no between- and within-group effects.

### Conclusion

Perturbation-based trunk stabilization training is possibly effective to improve the physical function of the lower back in elite rowers.

**Data Availability Statement:** Our preprint: https://osf.io/preprints/sportrxiv/6y9gm/ Supporting Information is available in our repository (https://doi.org/10.17605/osf.io/D983M) containing: • Detailed Intervention Description • Detailed results

• Processed and raw data • Footage of exercise •
Guideline checklists • Questionnaires.

**Funding:** This study was conducted within the
MiSpEx research network and funded by the
German Federal Institute for Sport Science (BISp)
[ZMVI1-080102A/11-18]. The funders had no role
in study design, data collection and analysis,
decision to publish, or preparation of the
manuscript.

**Competing interests:** The authors declare no
conflict of interest.

## Introduction

Low back pain (LBP) is a common health issue in elite athletes [1–3] and especially in rowing
[1, 4, 5]. Lifetime prevalence ranges between 65% [4] and 94–96% [2, 5]. The consequences are
high therapeutic costs, a breakdown of training and performance restrictions [6]. A state-of-
the-art breakdown for LBP prevalence, management and prevention in (sub-)elite rowers is
given by Wilson et al. [7] in their consensus statement.

Neuromuscular control plays a major role in trunk stability and LBP [8, 9] and deficits are
evident in LBP [10, 11]. Deep muscles (eg, M. Transversus Abdominis, M. Multifidus)–which
contribute essentially to trunk stability via segmental stabilization–showed a delay in muscle
activation after sudden perturbation in chronic LBP [11]. Human body movement involves
cascadic muscle activation from proximal to distal, even by just raising an arm [9, 12]. This
involves anticipatory (feedforward) muscle recruitment which relates to predictable perturba-
tions, whereas external perturbations rely on feedback mechanisms [9, 11, 13]. Neural cou-
pling suggests that trunk muscles are more likely to innervate together than distal muscles [8].
Nevertheless, precise control of deep muscles is important. Tsao et al. [14] demonstrated
altered deep muscle innervation by neuroplastic changes in the brain. In LBP, the representa-
tion locations of deep and superficial muscles in the motor cortex do not differ, whereas in a
matched healthy population they do. A possible compensation in LBP patients is increased
activation of superficial muscles to maintain trunk stability [8, 9]. This may lead to increased
loads for the spine architecture, which can be referred to as *tissue loading* [8]: Nociceptors con-
stantly transmit signals that increase pain via processes of sensitization/wind-up phenomenon
[8, 15]. Those ongoing nociceptive stimuli might increase the excitability of nerve cells that
might lead to hyperalgesia/allodynia [15, 16]. These maladaptive mechanisms might contrib-
ute to the persistence and chronification of LBP and ground the approaches of motor control
exercise [17].

Martinez-Valdes et al. [18] concluded an inefficient activation of the erector spinae in row-
ers with LBP due to higher activation of the erector spinae and a less complex EMG signal.
Further, rowing is characterized by high frequent bending of the lumbar spine [19], which pos-
sibly affects LBP or vice versa [20, 21]. Ergometer rowing seems to affect rowing kinematics by
more extensive flexion movements and therefore contributes to LBP [22, 23]. Additionally,
short-term fatigue appears to alter the rowing technique in the same manner and thus can
have detrimental effects in loading the spine [24, 25]. Supporting this hypothesis, rowers with
LBP seem to move their lumbar spine closer to end range flexion than healthy rowers [20].
When rowing to one side of the boat (sweep rowing) additional factors involve. Sweep rowing
is characterized by lateral flexion and rotation combined with (sagittal) flexion. Especially the
combination of rotation and flexion results in enhanced spinal strain [26]. Overall, these char-
acteristics contribute to the concept of tissue loading in rowers.

Perturbations lead to increased noise in the nervous system and invoke deep muscle con-
tractions. Moreover, neural networks formed under the presence of noise are more flexible in
handling external conditions [27]. Instability exercise is considered within the framework of
motor control exercise which effectiveness is evaluated in several meta-analyses [28–32]. How-
ever, the methods of instability exercise (including perturbations) and voluntary activation of
deep muscles (e.g. segmental stabilization exercise) differ. The nomenclature in the literature
is ambiguous: perturbation might be implicitly used in general stabilization exercise [28], but
there is a lack of specific research in perturbation-based trunk stabilization training (PTT).
Recently, a meta-analysis showed beneficial effects of PTT on pain and disability [33]. Another
study showed beneficial effects for athletes after 1 year of PTT for pain intensity, trunk exten-
sion/flexion strength [34].

We suggest that an intervention aiming to improve neuromuscular deficits could be beneficial for LBP in rowers. Therefore, we developed a tailored PTT based on the specific demands of the German national sweep rowing squad. We defined (back) pain intensity and disability as main outcomes. Further, trunk stability related measures were evaluated.

## Methods

### Study design

We conducted a 10-week prospective cohort study (parallel groups) 12 weeks were planned, but we rescheduled the post testing day based on training schedule adaptations. The main outcomes were measured one week before and after the intervention. The German national sweep rowing squad took part in this sport-specific intervention. Randomization was not feasible due to practical restrictions given by the coaches. Recruitment, diagnostics and intervention took place at the localities of the national team. The trial was retrospectively registered in the German Clinical Trial Register (DRKS00022264), conducted in agreement with the Declaration of Helsinki and approved by the local ethics committee of the Faculty of Sport Science, Ruhr University Bochum (EKS V 12/2018). We applied several guidelines to our report (CERT [35], TREND [36], CONSORT [37]).

We failed to pre-register this study because we were used to implementing training intervention studies without pre-registration. But per definition, interventions examining health-related outcomes are clinical trials and should be registered a priori. We, the authors, confirm that ongoing clinical trials will be registered before enrollment and like to refer the readers to the growing importance of pre-registration in light of the "replication crisis" and for the sake of transparent research [38].

### Participants

In April 2018, 36 of 37 male oarsmen of the German national sweep rowing squad (boat classes 8+, 4-, 4+, 2-) gave their written consent to participate in this study (Fig 1). 18 athletes from the actual national team were predetermined as intervention group who received perturbation-based trunk stabilization exercise (PTT) as treatment, while 18 junior athletes kept their usual routines as the control group (CG). Baseline tests started two weeks after recruiting.

### Data analysis

The inclusion criterium for data analysis was active participation (at least 12 of 16 sessions). Thus, 3 athletes were not included. 7 athletes dropped out in the qualification phase. Finally, data from 12 (PTT) and 14 (CG) athletes were included in the data analysis (dropout: 27.7%).

### Intervention

PTT was conducted conceptually involving sport-specific requirements. Therefore, we applied instability by internal and external perturbations to sport-specific movements and postures. Perturbations were applied by either rapid voluntary movements (internal perturbation) and unstable conditions, e.g. unstable surfaces, water-filled pipes, pushing from therapists (external perturbation). All of the total 16 group sessions were supervised by a physiotherapist and a sports scientist. The exercises were progressed by increasing levels of force and instability (Fig 2). The level was adapted by subjective rating of the supervisors and athletes and two numeric rating scales [1–10] of perceived exertion: 1) the CR-10 scale [39] rating perceived exertion from 1 to 10 and 2) an unvalidated instability scale with „1"defined as stable standing and „10"as maximal instability. The intensity of exercise was considered to progress when values

 

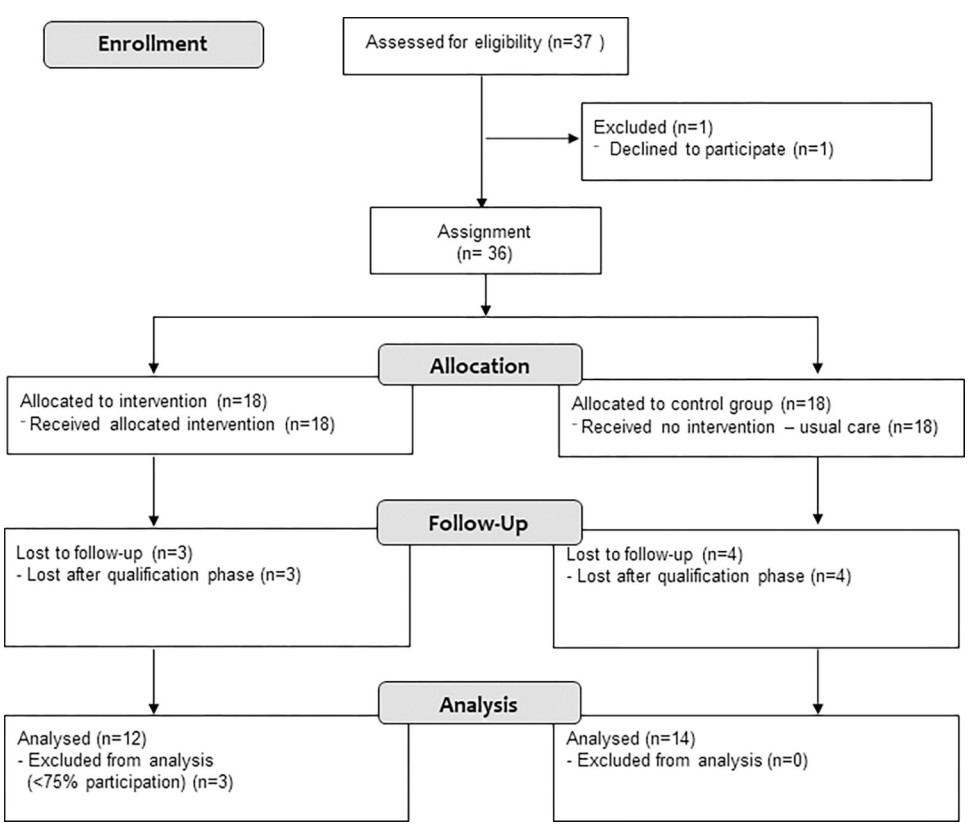

**Fig 1. CONSORT flow diagram.**

were lower than or equal to 5. 16 sessions á 30–40 min in 10 weeks were applied. A more detailed description according to the CERT guideline [35] can be found in our repository (S1).

## Diagnostics

Athletes were tested one week before and one week after the intervention. Additional measures of pain and disability were collected in weeks 4 and 7 of the training period. The test battery contained back pain questionnaires, measurements of maximal isometric trunk strength,

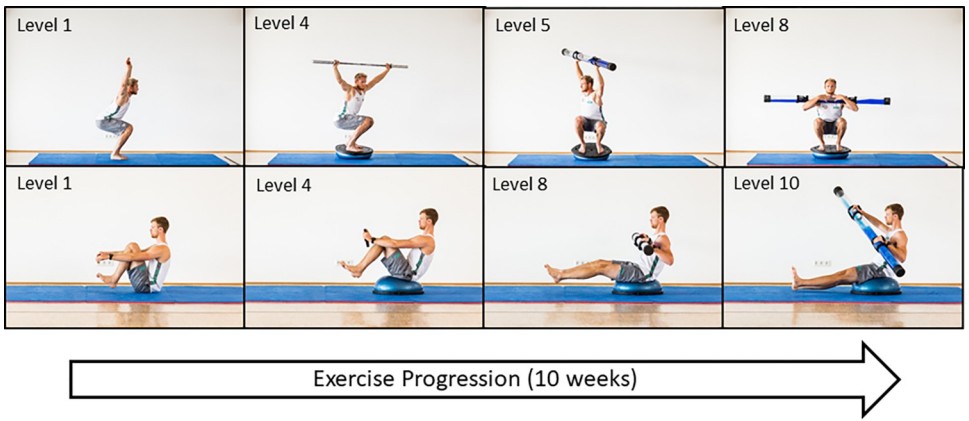

**Fig 2. Progression of the basic exercises squat (upper line) and rowing (lower line).**

counter movement and rebound jumps and stand stability. Participants were tested in small groups each day. We processed the data and calculated the outcomes of strength, jump and postural sway in Matlab (R2018b).

**Pain and disability.** We used two subscales from the 7-item Chronic Pain Grade Questionnaire by von Korff et al. [40]. Thus, the arithmetic means of items 1–3 and 4–6 were defined as pain intensity and disability, respectively. Those values were normalized to scale from 0 to 100. Reliability and Validity were examined for both the English [41] and the German versions [42]. Internal consistency was good (alpha = .88) for disability and moderate (alpha = .68) for pain; reliability for the whole instrument was good (alpha = .82) for the German version [42].

**Maximal isometric trunk strength.** Each participant performed three trials of maximal voluntary isometric trunk flexion and extension movements. Force was obtained using a strain gauge of 5 kN (KD80s, ME Meßsysteme GmbH, Hennigsdorf, Germany). For signal processing, a digital measurement amplifier (GSV-4BT M12, ME), a Bluetooth interface and the corresponding software (GSV Multichannel, ME) was used. The force-time curve was smoothened by a moving mean of 500 ms. The inter-trial rest was at least 3 minutes and the test time for each trial was approximately 5 seconds. Feet were placed on the ground, knees and hips fixed, sitting upright. A strap connected to the strain gauge and the apparatus was placed right under the armpits. To prevent peaks in the force curve, athletes were advised to preload the strap. Then, athletes applied maximal force with verbal encouragement from the instructor. Maximal force in Newton was obtained for each trial. Thus, the mean of the two best trials was defined as the outcome for each condition (flexion/extension).

Test-retest reliability and concurrent validity were tested by colleagues within our research group. Coefficient of variation was 2.0% for flexion and 5.0% for extension. Concurrent validity to isokinetic trunk measurement (Biodex 3 Medical System Inc., USA) was r = 0.73 for extension and r = 0.84 for flexion (n = 15).

**Counter movement jump.** Participants performed 8 standardized (hands gripping the hip, countdown), maximal counter movement jumps (CMJ) on a high-quality force plate (Kistler 9260AA6, Kistler Instrumente AG, Winterthur, Switzerland) after instruction and familiarization: two bipedal CMJ, two bipedal CMJ with rebound jump, one single-leg CMJ on each leg and one single-leg CMJ on each leg with rebound jump. The sequence of the first single-leg jump (left/right) was randomized between the participants. Each participant performed all jumps in the same order pre and post. A jump was repeated if the hands were loosened from the hip or a standing time of 3 s was not reached. Unfiltered data were used to calculate flight time $t$ based on the vertical force $F_z$. Hence, jump height $h$ was calculated with the transformed formula $h = gt^2/8$, where $g$ is the gravity constant. Plausibility was checked for every single jump by plotting $F_z$. The mean was calculated for bipedal jumps [43]. The results of the rebound jumps are not presented in this manuscript but the data are available in our repository (S2).

Exactly this setup showed excellent concurrent validity (Intra-Class-Correlation (2,1) > 0.999, Limits of Agreement -0.1 to 0.2 cm) compared to another high-quality force plate [44].

**Postural sway.** Participants performed a total of 3 stances (bipedal, single-leg left/right) over 30 s on a force plate (CSMi Computer Sports Medicine Inc., Stoughton, MA, USA) in a standardized manner (hands gripping the hip, barefoot, visual fixation in 2 meter distance). The sequence of the starting leg was randomized between the participants. Each participant performed all stances in the same order pre and post. The trace length of the centre of pressure was defined as the outcome of this measurement. Test-retest reliability of exactly this setup was good (ICC = 0.85). However, concurrent validity was moderate to good (0.49–0.83) with a systemic bias to lower values when compared to high-quality force plates [44].

## Minimal important change (MIC)

By evaluating the literature, cost and benefit [45–47], and our own data we set up a MIC for the between-group comparison of 5 points for the main outcomes CPI and DS. For individual changes, Ostelo et al. [47] proposed a 20 point threshold. We defined a MIC of 1 cm for jump height [48], and a MIC of 150 mm for single-leg stance [44]. The MIC of 70 N for strength outcomes is based on the standard error of measurement in the control group.

## Statistics

We chose to draw inference via an estimation approach rather than traditional (null) hypothesis testing [49–51]. We aimed to interpret practical relevance by contrasting confidence intervals to the region of 0 ± MIC [45, 52]. Therefore, we used 95% confidence intervals [lower bound, upper bound]. Point estimates on outcomes are presented as original units and standardized effect size (Hedge's g). Descriptive values are presented as means with standard deviation (SD).

Between-group effects were calculated via ANCOVA on pre-post change-scores with baseline adjustment to take account of individual differences and regression to the mean [53]; we also compared this to unadjusted change scores (ANOVA / t-test) for robustness. Thus, CI's were obtained from post-hoc procedures. Within-group effects for each group were estimated via marginal means of ANCOVA. To challenge the robustness of our findings, we carried out sensitivity analyses on our main outcomes (CPI, DS) by 1) analyzing the LBP subgroup defined by DS>0 and CPI>20 and 2) estimating the effect of most extreme single values by leave-one-out analysis. To estimate response heterogeneity, we compared the standard deviation of the change scores between groups [54, 55] and visually inspected scatter and violin plots. Lastly, relative evidence was evaluated by Bayes Factors (BF) from equivalent bayesian models with objective priors. Procedures were carried out in JASP (v 0.14) [56]. We evaluated the model residuals to check assumptions for normality (Q-Q plots) and homoscedasticity.

## Results

The groups differ considerably in age (5.6 years, $CI_{95\%}$ [3.5, 7.8]) and slightly in training volume (2.2 hours, [-2.3, 6.7]) due to the selection criteria (Table 1). No adverse events occurred during the supervised training period. The athletes completed 14.4 (1.4) of 16 sessions (n = 12) in 10 weeks. The mean training intensity was 5.3 (SD: 1.4, range: 3 to 7) rated on the CR-10 scale and the mean instability was 5.6 (SD: 1.4, range: 3 to 8). We observed considerable differences in the ratings within and between athletes, and between exercises. The mean differences between both scales range from -0.7 to 0.8 revealing varying demands of instability and overall strength of those exercises.

Table 2 shows the descriptive values of our outcomes.

**Table 1. Study group characteristics.**

| Variable | PTT (n = 12) | CG (n = 14) |
|---|---|---|
| Age [years] | 26.0 (3.7) | 20.4 (0.9) |
| Weight [kg] | 92.1 (11.2) | 89.7 (10.7) |
| Height [cm] | 192.2 (7.9) | 191.9 (7.0) |
| Training volume [h/week] | 24.3 (6.7) | 22.1 (4.4) |

All values are reported as mean (SD); PTT: perturbation-based trunk stabilization training, CG: control group

**Table 2. Outcome descriptive values.**

| Variable | Group | n | Pre | Post | Delta | $SD^2_{PTT}/SD^2_{CG}$ |
|---|---|---|---|---|---|---|
| Disability [0–100] | PTT | 12 | 11.4 (9.7) | 2.8 (5.3) | -8.6 (15.8) | 7.96 |
| | CG | 13 | 9 (8.9) | 7.3 (9) | -1.7 (5.6) | |
| Pain Intensity [0–100] | PTT | 12 | 33.9 (24.3) | 21.1 (13.0) | -12.8 (23.7) | 1.11 |
| | CG | 13 | 38.7 (22.1) | 21.9 (-4.6) | -16.8 (22.5) | |
| Trace sum [mm] | PTT | 12 | 2534 (412) | 2567 (504) | 33 (447) | 1.38 |
| | CG | 14 | 2507 (277) | 2496 (425) | -11 (380) | |
| Trace left [mm] | PTT | 12 | 1193 (213) | 1248 (50) | 55 (186) | 0.93 |
| | CG | 14 | 1288 (167) | 1234 (196) | -54 (193) | |
| Trace right [mm] | PTT | 12 | 1342 (250) | 1319 (288) | -23 (320) | 1.15 |
| | CG | 14 | 1219 (211) | 1262 (253) | 43 (299) | |
| Extension [N] | PTT | 11 | 1079 (263) | 1079 (196) | 0 (130) | 1.76 |
| | CG | 14 | 1154 (177) | 1107 (187) | -46 (98) | |
| Flexion [N] | PTT | 11 | 778 (138) | 785 (148) | 7 (76) | 0.84 |
| | CG | 13 | 755 (135) | 775 (117) | 20 (83) | |
| CMJ bipedal [cm] | PTT | 12 | 37.0 (4.3) | 35.2 (4.9) | -1.8 (3.2) | 0.94 |
| | CG | 14 | 36.4 (5.2) | 35 (4.2) | -1.4 (3.3) | |
| CMJ left [cm] | PTT | 11 | 15.4 (1.9) | 15.8 (1.8) | 0.4 (1.9) | 0.63 |
| | CG | 14 | 13.6 (2.4) | 14.4 (2.4) | 0.9 (2.4) | |
| CMJ right [cm] | PTT | 12 | 15.8 (2.2) | 15.5 (2.6) | -0.4 (2.6) | 2.64 |
| | CG | 14 | 14.6 (2.4) | 14.1 (2.5) | -0.5 (1.6) | |

All values are reported as mean (SD); PTT: perturbation-based trunk stabilization training, CG: control group, CMJ: counter movement jump, $SD_{PTT/CG}$: Standard deviation of change scores

The PTT group had less disability (5.3 points, $CI_{95\%}$ [0.4, 10.1], g = 0.42) than the control group after baseline adjustment (Fig 3). This effect remained stable in subgroup- and leave-one-out-analyses in adjusted (point estimates: 4.1 to 7.4) but less in unadjusted analysis (point estimates: 2.8 to 11.1). Furthermore, our data was less compatible with negative effects on disability (lower CI > lower MIC. The adjusted within-group change was -7.8 $CI_{95\%}$ [-11.2, -4.3] and -2.5 $CI_{95\%}$ [-5.8, 0.8] in PTT and CG, respectively.

Both groups experienced comparable reductions in pain intensity (Fig 3). The adjusted between-group effect in pain intensity was -1.0 points ($CI_{95\%}$ [-16.3, 14.3], g = -0.04). Change scores of pain intensity yield a bimodal in the PTT and a skewed distribution in the control group. Subgroup- and leave-one-out-analysis with baseline adjustment shifted effects towards benefits for PTT (point estimates: 1.1 to 9.0) challenging the robustness of this estimate. Nevertheless, CI's remained large yielding inconclusive results [45]. The adjusted within-group change was -14.4 points $CI_{95\%}$ [-25.4, -3.3] and -15.4 points $CI_{95\%}$ [-26, -4.8] in the PTT and the control group, respectively.

Between-group estimates for stand stability showed small effect sizes towards CG of -0.39, -0.07 and -0.12 (Hedge's g) for left stance, right stance and the sum, respectively (Fig 4). However, the data are compatible with a range of effects mostly covering the equivalence region. Adjusted and unadjusted point estimates (-53 to 55 mm) yield no substantial pre-post changes in both groups.

The between-group effects in trunk extension (-28 N $CI_{95\%}$ [−114, 58], g = -0.23) and flexion (8 N $CI_{95\%}$ [−57, 74], g = 0.09) showed substantial overlap with the equivalence region. Both adjusted and unadjusted pre-post changes indicate no substantial change for trunk strength in both groups (point estimates: -39–18 N, ½ CI width < 64 N).

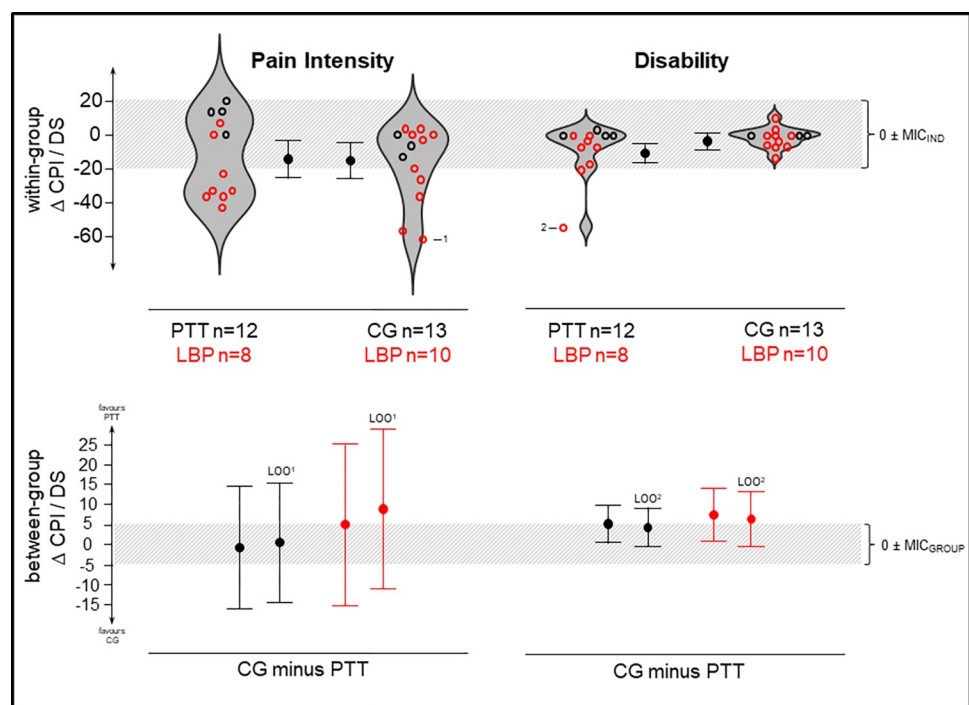

**Fig 3. Pain and disability outcomes.** Top plots show change scores in violin-scatter-plots accompanied by adjusted 95% CI obtained by marginal means of ANCOVA; bottom plots show adjusted between-group effects on both the whole experimental group (black) and the LBP subgroup (red)–positive values represent beneficial effects towards PTT (perturbation-based trunk stabilization training) over CG (control group); grey shaded: Null ± MIC (minimal important change) for group and individual changes; 1,2: data left out in leave-one-out-analysis; CPI: characteristic pain intensity, DS: disability score, LOO: leave-one-out analysis.

The point estimates for the between-group effect in all jumps were smaller than ½ MIC ($< 0.5$ cm) with standardized effect sizes of 0.08, -0.20 and 0.24 (Hedge's g) for bipedal, left and right jumps. However, the confidence interval was relatively wide compared to the equivalence region, yielding inconclusive results. Within both groups, there was a reduction in bipedal jump height (PTT: -1.7 cm $CI_{95\%}$ [-3.5, 0.1], CG: -1.5 $CI_{95\%}$ [-3.1, 0.2]) but not in single-leg jumps.

The overall strength of (relative) evidence was weak. For disability, the alternative hypothesis was two times more likely than under the null hypothesis (BF = 2). Weak evidence was

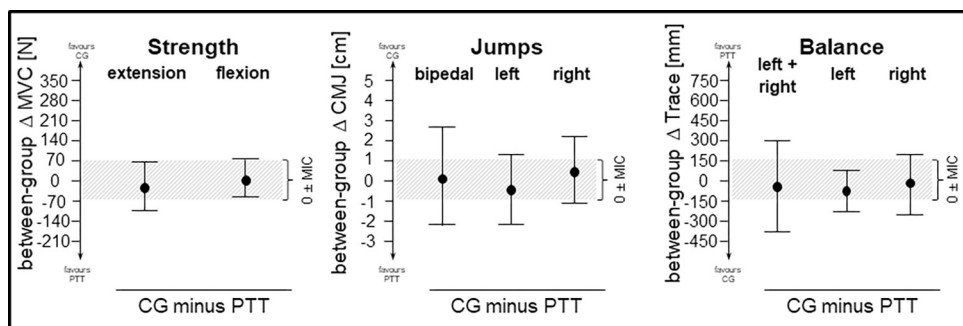

**Fig 4. Functional outcomes.** Plots show adjusted between-group effects of ANCOVA (95% CI); negative values in strength and jumps and positive values in balance represent beneficial effects towards PTT (perturbation-based trunk stabilization training) over CG (control group); grey shaded: Null ± MIC (minimal important change); MVC: maximal voluntary contraction, CMJ: counter movement jump.

found for the null hypothesis in pain intensity, right and summed trace, trunk strength, and jump height (BF = 2.2 to 2.7) and left trace (BF = 1.6).

## Discussion

We found a beneficial between-group effect for disability of 5 points towards PTT and equivalent or inconclusive results on pain intensity, trunk strength, jump height and stand stability.

Cochrane reviews found similar effects for disability improvements following motor control exercise in acute [31] and chronic [32] non-specific LBP. Perich et al. [57] found similar effects of a multi-dimensional study in female rowers. Thornton et al. [58] found a reduction of 2.6 points $CI_{95\%}$ [0, 5.1] in athletes in their meta-analysis. A larger sample of non-athletes was involved in a meta-analysis by Niederer et al. [33] which focuses specifically on PTT. They found a between-group effect of 4.8 points $CI_{95\%}$ [2.5, 7.2] for 12 weeks PTT with 2–3 sessions per week. Mean changes ranged from -15.5 to -1.6 within PTT in the underlying studies. Despite a lower session frequency (1.6 vs. 3) and duration (10 vs. 12 weeks), the compliance was slightly better in our study (90% vs. 66% attendance–unpublished data [59]). Nevertheless, we aimed to conduct two sessions per week, but we were restricted to the schedule of the athletes participating in international championships.

In this study, the low baseline values in disability might be confounding. We observed response heterogeneity by 8 times higher variance of change scores in the treatment group which might indicate worsening in a subgroup when small mean effects are present. However, sensitivity analyses (subgroup, leave-one-out) yield that 5 points with a margin of error of 5 points is a robust estimate given our data. Larger scale studies should be carried out to verify whether this result can be a true effect. Even small changes like 5 points on a 0–100 scale might be beneficial in elite athletes considering the relatively small amount of effort to conduct individualized PTT.

Observed pain intensity estimates were very uncertain and diverging in our sensitivity analyses. In the PTT meta-analysis, the between-group pain estimate was 4.3 points $CI_{95\%}$ [2.4, 6.1] [33]. Cochrane reviews declared similar effects for motor control exercise [31, 32]. The observed change within the PTT group stands in line with the meta-analysis [33], but the change in the control group was reasonably higher. One reason to explain these differences is small sample size bias; another would be a seasonal effect: In winter, rowers train more often on an ergometer which influences LBP [23]. As this study started in spring, LBP might have been reduced by reduced ergometer training volume. This link currently lacks evidence, we would need longitudinal observations of back pain in elite rowers to answer this question.

The distribution of change scores was bimodal in PTT and skewed in CG. It is discussed whether the distribution of pain associated measures follows a normal distribution [60]. However, the model residuals in ANCOVA's were normally distributed and no heteroscedasticity was found. Furthermore, two high delta values attributed to divergent results in our sensitivity analyses (e.g. leave-one-out). To address bimodality in the treatment group, Moore et al. [60] emphasize responder analysis to address response heterogeneity. Responder analyses should incorporate the control group considering common dichotomous methods are flawed [54, 61]. However, the variance of change scores yields no remarkable response heterogeneity, but assumptions of group-wise normality were violated [54]. To assure any conclusion, sophisticated methods [54, 61] might be used based on a higher sample size to address response heterogeneity.

Given our data, balance improvements in both groups were unlikely and the findings in the literature are divergent. Brachman et al. [62] reviewed studies on balance outcomes after related interventions in athletes. They concluded an effect but did not discuss the magnitude

due to different measurements (e.g. COP, stork, Star Excursion Balance Test, Y-Balance Test). Imai et al. [63] found a 9.8% decrease of postural sway (one-leg stance) in favour of trunk stability exercises in soccer players. Saunders et al. [64] found no difference in simple single-leg stance after severe balance training in figure skaters. Due to ceiling effects, Thompson et al. [65] concluded that more instable positions are necessary when testing athletes. Barbado et al. [66] demonstrated better trunk balance in kayakers while sitting and outlined the relevance of sport-specific test protocols. We like to emphasize this: Further research should incorporate specific balance measures (e.g. while sitting) to evaluate the efficacy of PTT or similar interventions on trunk stability given the conjunction to injury and low back pain [9, 62] and sport-specific performance.

Our data is compatible with between-group effects for strength less than 6% in extension and 8% flexion, respectively. This was not higher than the minimal important change defined by the typical error of the control group. We expected very small effects in elite rowers due to high baseline values. Sample size and intraindividual measurement noise restrict the certainty of our estimates. We conclude that the effect of PTT on maximal strength is negligible; however, PTT aims to address neuromuscular deficits rather than maximal trunk strength itself.

Outcomes on jump height yielded inconclusive results. Reviews showed that strength training did not increase jump height in elite rowers [67], but balance training in adolescents does [68]. The effect of PTT on jump height in elite rowers will remain unknown, but we hypothesize that single-leg jumps would profit more likely than bipedal jump performance due to task complexity. However, it is debatable whether the potentially small benefits of PTT on a complex task are worthwhile but might be of interest when discussing underlying mechanisms. While jump height decreased in both groups there might be an unknown confounder.

The observed effects are somewhat plausible considering the intended mechanisms. Perturbation invokes motor sensory neural pathways and therefore might inhibit noxious pathways [15]. We expected effects in postural control, but the small sample size and the test protocol might be confounding. We believe that overall effect sizes are small like in other one-size-fits-all interventions in non-specific low back pain. The superiority of exercise treatments by comparing means are hard to grasp [69]. Therefore, Hodges et al. [70] considered individual tailoring as a new research subject to enhance the effectiveness of treatments. Theoretical underpinnings of intended intervention mechanisms can be used for such tailoring. PTT might work differently than other interventions as outlined in the introduction. If we better understand the mechanisms, PTT might settle its position among many other treatments. Further research is necessary. To categorize PTT based on quantitative (e.g. EMG, accelerometers) and qualitative measures (e.g. instability scale) might help to further develop an exercise portfolio. Thereby, the mechanisms of unexpected (feedback) vs. expected (feedforward), as well as the severeness of perturbation should be evaluated. Lastly, in terms of precision medicine, repeated testing (e.g. replicated cross-over trials) might be underrated to evaluate inter- and intra-individual response among treatments [61].

## Practical application

The barriers to implementing PTT in elite athletes were low in our case. We observed high acceptance of our approach in elite rowers and positive feedback. Though this method is commonly used in many fields, the athletes in our study declared that many exercises were new to them. We adapted our approach in the very first sessions and included feedback from athletes and coaches. Despite the possible effectiveness, we verified the feasibility of individualized PTT in elite athletes.

## Limitations

Non-randomization [71] and small sample size are limitations of this study. Sampling size was restricted by studying a specific cohort: All sweep rowing athletes of the German National Team plus their back-ups. There is a considerably small population for which this sample might be representative. Our athletes successfully compete at the highest international level, but semi-professionals rowers might have similar training characteristics. Nevertheless, treatment response is probably small for pain and disability in LBP [70] and performance outcomes in high-level athletes due to ceiling effects. Thus, this study might be statistically underpowered for generalization purposes. Given the analyzed cases, we would have 80% power to detect standardized effect sizes greater than d = 1.15 with an alpha level of .05. To achieve the same for d = 0.5 one would need 51 (one-sided) or 64 (two-sided) subjects–including covariates might require less. Further, individualization may be a moderator. Before our study, athletes partly instructed themselves for trunk stabilizing exercise, thus, *how* the content was delivered might contribute to our estimates. Lastly, practitioners might set different boundaries for the minimal important change, which can yield different interpretations. We did not use 90% Confidence Intervals, which would reflect an alpha level of .05 for Equivalence Testing [45], but we did not employ strict hypothesis testing either.

## Conclusions

Perturbation-based trunk stabilization training is possibly effective to improve back pain related disability in elite rowers. Pain intensity decreased similarly in both groups, which might be caused by seasonal effects. Other outcomes tend to have no between- and within-group effects. However, the strength of evidence is small for these findings. Another study outcome is an exercise portfolio that can be used by practitioners, professional and recreative athletes to integrate perturbation-based exercises into their training routine. This training method is commonly used in the practical field but is yet to be fully researched.

## Supporting information

**S1 File. TREND statement checklist.**
(PDF)

**S2 File. CERT—Consensus on Exercise Reporting Template.** Study: Perturbation-based Trunk Stabilization Exercise in Elite Rowers.
(PDF)

**S3 File. 2017 CONSORT checklist of information to include when reporting a randomized trial assessing nonpharmacologic treatments (NPTs)**∗**.** Modifications of the extension appear in italics and blue.
(PDF)

**S4 File.**
(PDF)

**S5 File.**
(PDF)

## Acknowledgments

We like to thank the participating athletes and the coaching staff. Further, we like to thank our colleagues Christoph Schneider and Daniel Niederer for their feedback on early drafts of the manuscripts. We also like to thank all students who participated in data collection.

## Author Contributions

**Conceptualization:** Robin Schäfer, Hendrik Schäfer, Petra Platen.

**Data curation:** Robin Schäfer, Hendrik Schäfer.

**Formal analysis:** Robin Schäfer.

**Investigation:** Robin Schäfer, Hendrik Schäfer.

**Methodology:** Robin Schäfer, Hendrik Schäfer.

**Project administration:** Robin Schäfer, Petra Platen.

**Supervision:** Petra Platen.

**Visualization:** Robin Schäfer.

**Writing – original draft:** Robin Schäfer.

**Writing – review & editing:** Robin Schäfer, Hendrik Schäfer, Petra Platen.

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
