## [Decision Letter · Decision Letter 0]

18 Oct 2021

PONE-D-21-13623Perturbation-based Trunk Stabilization Training in Elite RowersPLOS ONE

Dear Dr. Schäfer,

Thank you for submitting your manuscript to PLOS ONE. After careful consideration, we feel that it has merit but does not fully meet PLOS ONE’s publication criteria as it currently stands. Therefore, we invite you to submit a revised version of the manuscript that addresses the points raised during the review process.

The manuscript has been evaluated by three reviewers, and their comments are available below. Two reviewers (including a statistical reviewer) have provided positive feedback on your manuscript. However, reviewer 3 has raised several concerns about the study design, methodology reporting and interpretation of the results.  Could you please revise the manuscript to carefully address the concerns raised?

We look forward to receiving your revised manuscript.

Kind regards,

Dario Ummarino, Ph.D.

Senior Editor

PLOS ONE

Journal Requirements:

2. Thank you for submitting your clinical trial to PLOS ONE and for providing the name of the registry and the registration number. The information in the registry entry suggests that your trial was registered after patient recruitment began. PLOS ONE strongly encourages authors to register all trials before recruiting the first participant in a study.

a) your reasons for your delay in registering this study (after enrolment of participants started);

b) confirmation that all related trials are registered by stating: “The authors confirm that all ongoing and related trials for this drug/intervention are registered

4. Please ensure that you refer to Figure 1 in your text as, if accepted, production will need this reference to link the reader to the figure.

5. We note that Figure 2 in your submission contain copyrighted images. All PLOS content is published under the Creative Commons Attribution License (CC BY 4.0), which means that the manuscript, images, and Supporting Information files will be freely available online, and any third party is permitted to access, download, copy, distribute, and use these materials in any way, even commercially, with proper attribution. For more information, see our copyright guidelines: http://journals.plos.org/plosone/s/licenses-and-copyright.

7. We note that Figure 2 includes an image of a participant in the study. 

Reviewers' comments:

Reviewer's Responses to Questions

**Comments to the Author**

1. Is the manuscript technically sound, and do the data support the conclusions?

Reviewer #1: Yes

Reviewer #2: Yes

Reviewer #3: No

2. Has the statistical analysis been performed appropriately and rigorously? 

Reviewer #1: Yes

Reviewer #2: Yes

Reviewer #3: Yes

3. Have the authors made all data underlying the findings in their manuscript fully available?

Reviewer #1: Yes

Reviewer #2: Yes

Reviewer #3: Yes

4. Is the manuscript presented in an intelligible fashion and written in standard English?

Reviewer #1: Yes

Reviewer #2: Yes

Reviewer #3: Yes

5. Review Comments to the Author

Reviewer #1: A two-arm controlled study was conducted to investigate the effects of perturbation-based trunk stabilization training (PTT) on back pain intensity and disability, maximum isometric trunk extension and flexion, jump height and postural sway of single-leg stance in male members of a German rowing team. Less disability was observed in the PTT group than in controls. Similar decreases in pain were observed in both groups.

Minor revisions:

1- In the abstracts, indicate that the study was non-randomized.

2- Prior to applying the ANCOVA, indicate if the distribution of the data checked for normality.

3- Since the study was not randomized, provide p-values to compare the baseline characteristics shown in Table 1.

4- The standard statistical terminology for “average” is “mean.”

Reviewer #2: Dear all

I realize that authors have many journals to consider when they want to publish their work, so I appreciate your interest in PLOS ONE; I am very happy to be able to write in a positive way.

It is evident that you have put a great deal of effort into this project and I want to praise your efforts,

The actual contribution from your study is clear and strong. The manuscript as currently written suggests that it might be suitable for sharing information about this field, and the data that you reported are representative to state with certainty your conclusions.

I should like to thank you for give me an opportunity to consider this work for publication. Great paper.

There is an error at introduction section at fourth line, replace low back pain with LBP.

Best Regards

Reviewer #3: Thank you for the opportunity to review this paper. General Comments:

Unfortunately I think the paper has some challenges. A significant number of these have been eluded to by the authors in the limitations section but they are issues that require addressing before publication.

in particular the study is not as robust as it could because, a) the participants were not randomly allocated to the intervention and control group, b) the study is underpowered c) the power analysis does indicate what the required number would be to potentially reach significance but this was indicated retrospectively d) the results do not show any real meaningful differences between the groups so this does not really add to the body of knowledge in this area. Perhaps the paper could be reworked as a pilot study then the results could be used to inform a more robust design that tests to see if the perturbation training actually is effective. The paper could also do with some review of the written English in places.

Specific Comments

Title: I wonder if a better title might be Perturbation based trunk motor control in elite rowers. The main construct of the paper is on motor control rather than stabilisation. Stabilisation is a more challenging term and implies changing an unstable situation. These rowers need to show greater control of trunk flexion as indicated in some of the introduction sections

Introduction

line 34 insert the word the before Consequences

Line 35 insert the word a before breakdown

Line 35 reword for example, rowers missed training to with rowers often missing training. This first paragraph is an example of three changes in English that improve the readability. Please look at other improvements where noted in the paper

This whole first section of low back pain in rowing needs to be situated in the terms on the latest consensus statement on LBP and rowing and refer to this as rowing related LBP see Wilson, F., Thornton, J. S., Wilkie, K., Hartvigsen, J., Vinther, A., Ackerman, K. E., Caneiro, J. P., Trease, L., Nugent, F., Gissane, C., McDonnell, S. J., McGregor, A., Newlands, C., & Ardern, C. L. (2021, Mar 8). 2021 consensus statement for preventing and managing low back pain in elite and subelite adult rowers. Br J Sports Med. https://doi.org/10.1136/bjsports-2020-103385

I think there are also other references that could be used around the incidence and prevalence of RR LBP see

Newlands, C., Reid, D., & Parmar, P. (2015, Jul). The prevalence, incidence and severity of low back pain among international-level rowers. Br J Sports Med, 49(14), 951-956. https://doi.org/10.1136/bjsports-2014-093889

Line 38 what specific deep muscles are you referring to, needs to be clear

remove the hyphen after deep muscles

line 39 needs rewording it seems unclear do you mean showed delay in muscle activation with perturbations?

Line 44 has this precise control been shown to be required in rowing, needs to be contextualised

line 69 Not clear what this different approach is

Line 71 what were the findings of the meta analysis referred to?

line 72 Another study evaluated PTT in high-level athletes.... what was the result, Not clear and not referenced

line 74. you suggest the PPT could improve neuromuscular deficits in rowers. Going back to my comment about the title, it may be better to make a clear link between changing any potential deficits and the ability to influence or improve motor control rather than trunk stability

Study design

line 82 as stated before no randomisation is a flaw and as such the current model may have introduced bias in the selection

Participants

Line 92 More detail is required on what the usual routines of the CG were to make this reproducible

Data Analysis

A power analysis should have been done here to determine the number of participants

Intervention

line 106 was it any or all of the session that were supervised. What are sports therapists? are these physiotherapists or strength and conditioning trainers?

Line 171 what is the significance of the reference to cost benefit as you did not measure this

Statistics

This is quite a detailed section but I am not clear why p values are not presented following tests like t test and ancova as this would be the usual convention and easier for the reader to follow.

Results

Were t test performed on the demographic data? You state in line 197 The groups differ considerably in age and slightly in training volume but no p values are reported to indicate significance

Discussion

Line 301 you state Further research should incorporate specific balance measures (e.g. while sitting) to

302 evaluate the efficacy of PTT or similar interventions on trunk stability given the conjunction to injury

303 and low back pain [9,59] and sport-specific performance. Why was this not considered from the start as these seems key to the whole study

line 308 you state PTT aims to address neuromuscular deficits rather than maximal trunk strength itself. So why measure strength and not changes in motor control as suggested with the title change

line 312 remove the word guess, non scientific language and 313 discussable?? if this a word

line 324 you state PTT might work differently than other interventions, how might this be different, please expand

Practical Application

line 333 can you clarify what a usual trunk stabilization training routine is. This is a construct not clear in the paper

line 334 you state We observed high acceptance of our approach in elite rowers and positive feedback. Though this method is commonly used in many fields, the athletes in our study declared that many exercises were new to them. Where did this information come from? Not in the results

Limitations.

Apart from all the key ones mentioned in the start what does this mean line 350 athletes partly instructed themselves for

trunk stabilizing exercise, this would seem a major confounder to the whole study!

Conclusion

You state this PTT is possibly effective. I dont think you can state this as no meaningful difference was really found between the groups.

Also line 358 you state Pain... which might be caused by seasonal effects. What does this mean ??

6. PLOS authors have the option to publish the peer review history of their article (what does this mean?). If published, this will include your full peer review and any attached files.

Reviewer #1: No

Reviewer #2: No

Reviewer #3: No

---

## [Author Response · Author response to Decision Letter 0]

19 Nov 2021

Dear Editor and Reviewers,

thank you for your work! We submitted our response to the comments raised in the rebuttal letter.

---

## [Decision Letter · Decision Letter 1]

6 May 2022

Perturbation-based Trunk Stabilization Training in Elite Rowers

PONE-D-21-13623R1

Dear Dr. Schäfer,

We’re pleased to inform you that your manuscript has been judged scientifically suitable for publication and will be formally accepted for publication once it meets all outstanding technical requirements. As you can see from the comments included underneath my signature below, the reviewers are unanimously satisfied that their previous concerns have been adequately addressed. Please note that Reviewer 3 has provided some additional minor recommendations, and I invite you to carefully consider these when your manuscript is returned for final technical checks.

Kind regards,

Dario Ummarino, Ph.D.

Senior Editor

PLOS ONE

Additional Editor Comments (optional):

Reviewers' comments:

Reviewer's Responses to Questions

**Comments to the Author**

1. If the authors have adequately addressed your comments raised in a previous round of review and you feel that this manuscript is now acceptable for publication, you may indicate that here to bypass the “Comments to the Author” section, enter your conflict of interest statement in the “Confidential to Editor” section, and submit your "Accept" recommendation.

Reviewer #1: All comments have been addressed

Reviewer #2: All comments have been addressed

Reviewer #3: (No Response)

2. Is the manuscript technically sound, and do the data support the conclusions?

Reviewer #1: (No Response)

Reviewer #2: Yes

Reviewer #3: Yes

3. Has the statistical analysis been performed appropriately and rigorously? 

Reviewer #1: (No Response)

Reviewer #2: Yes

Reviewer #3: Yes

4. Have the authors made all data underlying the findings in their manuscript fully available?

Reviewer #1: (No Response)

Reviewer #2: Yes

Reviewer #3: Yes

5. Is the manuscript presented in an intelligible fashion and written in standard English?

Reviewer #1: (No Response)

Reviewer #2: Yes

Reviewer #3: Yes

6. Review Comments to the Author

Reviewer #1: (No Response)

Reviewer #2: Dear Authors

I should like to thank you for give me an opportunity to consider this work for publication. You well done the a point by point answer to the comments of the reviewers.

Reviewer #3: Thank you to the authors for responding to my comments. For the most part they have addressed the issues raised and I am happy with the explanation provided. The title is reflective of the work but given the small numbers I would still recommend calling this a pilot study. I understand the challenges of using a small pool of elite athletes and the results are still useful but it is still an non randomised study and small sample size. As the findings of this research move forward a more robust study may be developed, hence pilot is more reflective if where this study currently sits

There are couple of minor typos and clarifications to be addressed

Abstract Line 18, Can Physical Therapists and Sports Therapist be added to reflect the changes later in text

Line 25 consider removing tend to and just say had no differences

Introduction Line 42 Can you please look at the clarity of this sentence Do you mean movement of the trunk with arm movement?

Line 73 reword muscles quite differ. to muscles are thought to differ

7. PLOS authors have the option to publish the peer review history of their article (what does this mean?). If published, this will include your full peer review and any attached files.

Reviewer #1: No

Reviewer #2: No

Reviewer #3: **Yes: **Duncan Arthur Reid

---

## [Editor Report · Acceptance letter]

11 May 2022

PONE-D-21-13623R1 

Perturbation-based Trunk Stabilization Training in Elite Rowers: a Pilot Study 

Dear Dr. Schäfer:

I'm pleased to inform you that your manuscript has been deemed suitable for publication in PLOS ONE. Congratulations! Your manuscript is now with our production department. 

Kind regards, 

on behalf of

Dr Dario Ummarino, PhD 

Staff Editor

PLOS ONE